# SLIM: STRUCTURE-AWARE LOW-RANK INFERENCE MODEL

## ABSTRACT

This paper introduces a new method for the low-rank compression of large language models. Existing techniques typically compress the weights individually, overlooking the internal dependencies within a transformer block. To address this limitation, we formulate a joint optimization problem to find the optimal low-rank weights for an entire transformer block, thereby minimizing the output reconstruction error. Our formulation allows the incorporation of key architectural elements, including residual connections and normalizations. We then introduce SLIM, an efficient algorithm to solve this optimization problem. Experimental results demonstrate that our method consistently achieves task accuracy improvements of over 5% compared to existing techniques across a range of compression ratios and model families.

## 1 INTRODUCTION

The size of large language models (LLMs) has grown rapidly in recent years, requiring substantial computational and memory resources for deployment. To mitigate these costs, LLM compression has become an effective strategy to reduce model size with minimal overhead. In this paper, we focus on low-rank compression, a technique that approximates an original weight matrix with two smaller low-rank matrices to save both memory and computation without requiring special hardware.

The most straightforward low-rank compression method is to apply truncated SVD to each weight matrix. While this technique has shown promise on earlier models (Shim et al., 2017; Sindhwani et al., 2015; Yu et al., 2017), it performs poorly on modern neural architectures, where weight matrices are often not inherently low-rank (Chen et al., 2021; Hsu et al., 2022). To address this limitation, several methods have recently been proposed that incorporate different importance weights into SVD. For instance, FWSVD (Hsu et al., 2022) uses the fisher information matrix to guide the decomposition, while ASVD (Yuan et al., 2023) leverages the norms of the rows of the empirical covariance matrix. Both DRONE (Chen et al., 2021) and SVD-LLM (Wang et al., 2025) utilize the full empirical covariance matrix to determine importance. These methods guide the model compression to be more accurate in the denser subspace of the inputs and improves the performance.

However, despite improving over vanilla SVD, these approaches still treat each weight matrix in isolation. In this paper, we argue that effective low-rank compression should be formulated at the transformer block level rather than at the level of individual matrices. By casting it as a joint optimization problem that minimizes the mean squared error of the entire block, we can naturally account for architectural components such as residual connections and RMS normalization. Although this joint objective does not admit a closed-form solution, it provides a more principled treatment of the block structure. We then propose a greedy scheme that sequentially updates each weight matrix, where each step reduces to solving a regularized rank-constrained linear regression problem with an efficient closed-form solution. Furthermore, we show that a gradient-descent–based refinement can further enhance the performance of our low-rank compression method.

Our contributions can be summarized as follows:

- We propose a novel formulation for jointly optimizing the low-rank approximation of transformer blocks, explicitly accounting for architectural elements such as residual connections and RMS normalization. We develop a model compression algorithm based on approximately solving this joint optimization problem.

- Extensive experiments across multiple LLMs demonstrate the effectiveness of our method. Under the same parameter constraint, our approach improves average classification accuracy by more than $5\%$ across six benchmarks compared to existing methods. Furthermore, we show that our method is compatible with other techniques such as adaptive rank selection and quantization.

## 2 BACKGROUND

### 2.1 STRUCTURE OF TRANSFORMER-BASED LLMS

Prevailing Large language models (LLMs) are mostly transformer models, which consists of a sequence of $L$ Transformer layers, denoted as $f^1, \ldots, f^L$. Each Transformer layer consists of two sub-blocks with residual streams: the Attention block and the Feed-Forward Network (FFN) block.

Consider the input hidden states of the $l$-th Transformer layer, denoted as $X^l \in \mathbb{R}^{d \times n}$, where $n$ is the number of samples[1] and $d$ is the model dimensionality, the forward pass can be written as follows:

$$f^l_{\text{Attention}}(X^l) = \text{Attention}\big(X^l; W^l_{\text{Attention}}\big) + X^l, \tag{1}$$

$$f^l_{\text{FFN}}(X^l) = \text{FFN}\big(f^l_{\text{Attention}}(X^l); W^l_{\text{FFN}}\big) + f^l_{\text{Attention}}(X^l). \tag{2}$$

In $W^l_{\text{Attention}}$, it usually contains the $W_{\text{query}}, W_{\text{key}}, W_{\text{value}}$ weight matrices, which are responsible for computing the query, key, and value embeddings respectively. The final weight matrix $W_{\text{output}}$ then projects the attention output back to the original hidden space. In $W^l_{\text{FFN}}$, there can be multiple weight matrices, while there is a final down projection matrix $W_{\text{down}}$ that maps the FFN output back to the original hidden space.

Residual connections, as shown in equations equation 1 and equation 2, are crucial to the performance of transformer models. Moreover, within both the Attention and Feed-Forward Network (FFN) blocks, normalization layers play a critical role. These layers can be applied either at the beginning (pre-layer normalization) or at the end of the block (post-layer normalization). Among the various normalization techniques, Root-Mean-Square normalization (RMSNorm), introduced by Zhang & Sennrich (2019), has become the most widely adopted.

### 2.2 LOW-RANK LLM COMPRESSION

The goal of low-rank LLM compression is to approximate each weight matrix with a low-rank counterpart. When considering a weight matrix $W^l_h$ of layer $l$ in isolation, its optimal low-rank approximation is given by truncated Singular Value Decomposition (SVD):

$$\bar{W}^l_h = \text{SVD}_r(W^l_h), \tag{3}$$

where $r$ is the target rank. The number of parameters and flops can then be reduced from $O(mn)$ to $O(r(m+n))$.

However, this SVD-based compression ignores the distribution of the inputs, $X^l_h$, to these weight matrices. Intuitively, compression should be more accurate in the denser subspace of the inputs and can be less precise in sparser regions. Motivated by this, several existing low-rank compression methods utilize a calibration set to obtain the empirical distribution of $X^l_h$. This allows them to perform a more optimal, data-aware compression that accounts for the input distribution (Hsu et al., 2022; Yuan et al., 2023; Chen et al., 2021; Wang et al., 2025).

## 3 PROPOSED METHOD

Instead of compressing each layer in isolation, we argue that an effective compression algorithm should consider the entire Transformer sub-block jointly. For each sub-block $f^\ell$, we propose that one should be solving the block MSE objective:

$$\min_{\bar{f}^l \in \Pi} \|\bar{f}^l(\bar{X}^l) - f^l(X^l)\|_F^2, \tag{4}$$

---

[1] In practice, its amounts to number of tokens, which is batch size times sequence length

where $X^l$ and $\bar{X}^l$ are the inputs to the original and compressed sub-blocks, respectively. The set $\Pi$ represents all compressed sub-blocks $\bar{f}^l$ where each weight matrix adheres to its specified rank constraint.

For general $f^l$, there is no closed-form solution for equation 4. We therefore adopt a greedy strategy that sequentially minimizes the output error of each weight matrix. Specifically, we define a sequence of sublayers $h$, each parameterized by a single, newly uncompressed weight matrix $W_h$, in conjunction with a selection of previously compressed matrices. For each sublayer, we then solve for its compressed counterpart $\bar{h}$ by optimizing:

$$\min_{W_{\bar{h}} \in \Pi_r} \|\bar{h}(\bar{X}^l) - h(X^l)\|_F^2, \tag{5}$$

where $\Pi_r$ is the low-rank constraint. For most weight matrices $W_h$, we can simplify the sublayer to a linear mapping $h(X^l) = W_h g(X^l)$, where $g(X^l)$ is the layer's input. Therefore, compressing $W_h$ reduces to solve the following least-squares problem:

$$\min_{W_{\bar{h}} \in \Pi_r} \|\bar{W}_{\bar{h}} \bar{g}(\bar{X}^l) - W_h g(X^l)\|_F^2. \tag{6}$$

However, two common structures in LLMs—residual connections and normalization—require special consideration, which we detail below.

**Residual Connection.** Residual connection (He et al., 2016) has been commonly used in LLMs, including Mistral (Jiang et al., 2024), Gemma (GemmaTeam, 2025), Qwen3 (Yang et al., 2025), and more. Let $h$ be a sublayer of $f^l$ satisfies the form of

$$h(X^l) = r_{\text{res}}(X^l) + W_h g(X^l),$$

where $r_{\text{res}}$ is the residual connection. Plugging this into our objective in equation 5 gives

$$\min_{W_{\bar{h}} \in \Pi_r} \|\bar{r}_{\text{res}}(\bar{X}^l) + \bar{W}_{\bar{h}} \bar{g}(\bar{X}^l) - r_{\text{res}}(X^l) - W_h g(X^l)\|_F^2. \tag{7}$$

Compared to equation 6, this objective introduces an extra term $\bar{r}_{\text{res}}(\bar{X}^l) - r_{\text{res}}(X^l)$, which we call the residual connection. This term becomes significant as the compression error accumulates across the transformer blocks. This structure appears in pre-layer normalization transformers for the output projection matrix $W_{\text{output}}$ of the Attention layer and the final projection matrix $W_{\text{down}}$ of the FFN layer.

**Normalization.** Root mean square normalization (RMSNorm) has been widely used in recent transformer architectures (GemmaTeam, 2025; Yang et al., 2025; Bi et al., 2024). Let $h$ be a sublayer of $f^l$ satisfies the form of

$$h(X^l) = \text{RMSNorm}(W_h g(X^l)) \equiv W_h g(X^l) / \|W_h g(X^l)\|.$$

Plugging this into our objective in equation 5 gives

$$\min_{W_{\bar{h}} \in \Pi_r} \|(\bar{W}_{\bar{h}} \bar{g}(\bar{X}^l) / \|\bar{W}_{\bar{h}} \bar{g}(\bar{X}^l)\|) - (W_h g(X^l) / \|W_h g(X^l)\|)\|_F^2. \tag{8}$$

Compared to equation 6, this formulation includes an additional normalization by the RMS norm. This normalization is crucial in many cases; without it, the RMSNorm layer will degenerate to a simple scaling factor. This structure appears in post-layer normalization transformers like Gemma-3 (GemmaTeam, 2025) for the final projection matrix $W_{\text{output}}$ of the Attention layer and $W_{\text{down}}$ of the FFN layer, or for the query projection matrix $W_{\text{query}}$ and the key projection matrix $W_{\text{key}}$ for transformers that adopt Query-Key Normalization (QK-norm) like Gemma-3 (GemmaTeam, 2025) and Qwen-3 (Yang et al., 2025).

**Regularization** Additionally, we observe that it is important to add regularization to avoid overfitting to the sampled set. With this, equation 6, 7, and 8 with regularization can be formulated as the following reduced-rank linear regression problem

$$\min_{W_{\bar{h}}; \text{rank}(\bar{W}_{\bar{h}}) \leq r} \|\bar{W}_{\bar{h}} X - Y\|_F^2 + \eta \|\bar{W}_{\bar{h}} - W_h\|_F^2, \tag{9}$$

where $\eta > 0$ is a hyperparameter that controls the regularization strength.

For equation 6, we have $X = \bar{g}(\bar{X}^l)$ and $Y = W_h g(X^l)$. Similarly, equation 7 can be written as equation 9 with $X = \bar{g}(\bar{X}^l)$ and $Y = W_h g(X^l) + r_{\text{res}}(X^l) - \bar{r}_{\text{res}}(\bar{X}^l)$.

For the RMSNorm case (equation 8), under the approximation

$$\|\bar{W}_{\bar{h}} \bar{g}(\bar{X}^l)\| \approx \|W_h g(X^l)\|,$$

we have $X = \bar{g}(\bar{X}^l)/c$ and $Y = W_h g(X^l)/c$, where $c = \|W_h g(X^l)\|$.

Our choice of regularization penalizes the deviation from the original weights. We observe that this performs better than the standard $\ell_2$ regularization

$$\min_{\bar{W}_{\bar{h}};\text{rank}(\bar{W}_{\bar{h}})\leq r} \|\bar{W}_{\bar{h}} X - Y\|_F^2 + \eta\|\bar{W}_{\bar{h}}\|_F^2, \tag{10}$$

which unnecessarily suppresses parts of $\bar{W}_{\bar{h}}$ that also appears in $W_h$. To match the scale of $X$, we use $\eta = \eta_0 \|X\|$.

**Closed Form Solution**  We show that the closed-form of the above formulation is given by the following theorem.

**Theorem 1** *The optimal solution to equation 9 is*

$$\bar{W}_{\bar{h}} = \text{SVD}_r((\hat{Y}\hat{X}^\dagger)\hat{X})\hat{X}^\dagger, \tag{11}$$

*where $\hat{X} = \text{concat}([X, \eta^{1/2}I]), \hat{Y} = \text{concat}([Y, \eta^{1/2}I])$ respectively concatenates $X$ and $Y$ with an identity matrix along the second dimension, and $\hat{X}^\dagger$ is the pseudo-inverse of a matrix $\hat{X}$.*

The proof is in the appendix. This result is an extension of the reduced-rank linear regression solution developed in Izenman (1975), where they derived the closed form solution of $\min_{\bar{W}_{\bar{h}};\text{rank}(\bar{W}_{\bar{h}})\leq r} \|\bar{W}_{\bar{h}} X - Y\|_F^2$ without the regularization term.

**Complexity**  For the efficient computation of equation 11, let $S = (\hat{X}\hat{X}^T)^{-1/2}$ be the inverse of the positive semi-definite matrix square of the empirical covariance matrix $\hat{X}\hat{X}^T$, we have

$$\text{SVD}_r(\hat{Y}\hat{X}^\dagger\hat{X})\hat{X}^\dagger = \text{SVD}_r(\hat{Y}\hat{X}^T(\hat{X}^\dagger)^T)\hat{X}^\dagger = \text{SVD}_r(\hat{Y}\hat{X}^T S)S.$$

The computation time of $S$ is $O(nd^2 + d^3)$. Without loss of generality, assuming the output dimension of $Y$ to be a constant multiple of $d$, then the computation time of $\hat{Y}\hat{X}^T S$ is also $O(nd^2 + d^3)$, while computing its SVD takes $O(d^3)$. Consequently, the total time complexity for SLIM is $O(nd^2 + d^3 + O_{\text{forward}})$, where $O_{\text{forward}}$ is the complexity to compute $X$ and $Y$. For memory complexity, it is worth noticing that one can compute $\hat{X}\hat{X}^T$ and $\hat{Y}\hat{X}^T$ streamingly without storing the entire $\hat{X}$ and $\hat{Y}$ in the main memory, thus the complexity is $O(d^2)$. These complexities are identical to SVD-LLM (Wang et al., 2025).

Algorithm 1 summarizes our proposed method.

## 4 RELATED WORK

**Low-rank Compression.**  Low-rank compression is a widely-used technique for model compression which approximates a weight matrix with two smaller low-rank matrices. To obtain these matrices, Singular Value Decomposition (SVD) is the most common mathematical tool. Beyond simple SVD, several existing methods utilize a calibration set to obtain an empirical distribution of input activations, $X^l$, and compute importance weights to guide compression. FWSVD (Hsu et al., 2022) leverages the Fisher Information Matrix to determine the importance of individual parameters. ASVD (Yuan et al., 2023) computes a diagonal matrix based on the norms of the rows of the empirical covariance matrix of the input activations $X^l$. DRONE (Chen et al., 2021) minimizes the MSE error of the matrix output on the input activations while utilizing the entire empirical covariance matrix. Their experiments focus on the compression of BERT models. SVD-LLM (Wang

---

**Algorithm 1 SLIM**

---

1: **for** each $W_h$ sorted by the order of computation **do**
2:     Obtain the input $g(X^l)$
3:     Obtain the target rank $r$ for the compressed weight $\bar{W}_h$
4:     **if** the output of $W_h$ is normalized by RMSNorm **then**
5:         # e.g. $W_{\text{query}}$, $W_{\text{key}}$ with QK-norm, and $W_{\text{output}}$, $W_{\text{down}}$ for post-layer norm LLMs
6:         Set $X = \bar{g}(\bar{X}^l)/\|W_h g(X^l)\|$ and $Y = W_h g(X^l)/\|W_h g(X^l)\|$
7:     **else if** the output of $W_h$ is added with a residual $r_{\text{res}}(X^l)$ **then**
8:         # e.g. $W_{\text{output}}$, $W_{\text{down}}$ for pre-layer norm LLMs
9:         Set $X = \bar{g}(\bar{X}^l)$ and $Y = W_h g(X^l) + r_{\text{res}}(X^l) - \bar{r}_{\text{res}}(\bar{X}^l)$
10:     **else**
11:         Set $X = \bar{g}(\bar{X}^l)$ and $Y = W_h g(X^l)$
12:     **end if**
13:     Compute $\hat{X} = \text{concat}([X, \eta^{1/2}I])$ and $\hat{Y} = \text{concat}([Y, \eta^{1/2}I])$
14:     Compute $S = (\hat{X}\hat{X}^T)^{-1/2}$
15:     Obtain the compressed weight $\bar{W}_{\bar{h}} = \text{SVD}_r(\hat{Y}\hat{X}^T S)S$
16: **end for**

---

et al., 2025) also minimizes the MSE error of the matrix output and they extend the experiments to modern-day LLMs.

A key limitation of these methods is their failure to account for the propagation of compression error through the network. They operate by compressing each weight matrix using the activations from the full, uncompressed model, ignoring the cumulative effect of cascading errors. In contrast, our proposed block MSE objective considers the compressed input $\bar{X}^l$ for each transformer block and optimizes the compression of the entire block as a unit. Additionally, these existing methods treat each weight matrix in isolation as a simple matrix multiplication, overlooking critical architectural features found in modern transformer blocks, such as residual connections and Root Mean Square (RMS) normalization. Our paper demonstrates the importance of incorporating these structures into the compression for superior performance.

**Other Compression Methods.**    Besides low-rank compression, other compression methods fall into two categories: pruning and quantization. LLMPruner (Ma et al., 2023) considers gradient information to identify inter-dependent structures, which then prunes the least important parameter groups accordingly. BlockPruner (Zhong et al., 2024) assesses the importance (via PPL) of Attention and MLP blocks in the LLMs and applies a heuristic for iterative pruning; On the other hand, quantization methods (Dettmers et al., 2023; Lin et al., 2024) reduce the precision of weight matrices or input activations. Our proposed method is complementary to quantization techniques, as shown in Table 6.

## 5 EXPERIMENTS

### 5.1 SETUP

**Comparing Methods.**    We compare our proposed method with state-of-the-art low rank compression methods: FWSVD (Hsu et al., 2022), ASVD (Yuan et al., 2023), SVD-LLM (Wang et al., 2025), and AdaSVD (Li et al., 2025), along with other model pruning methods, including LLM-Pruner (Ma et al., 2023) and BlockPruner (Zhong et al., 2024).

In this paper, we focus on achieving a better low-rank approximation with a given parameter ratio constraint. However, among the low-rank compression baselines, ASVD, SVD-LLM, and AdaSVD include mechanisms to adaptively assign different parameter ratios to each matrix. To demonstrate the potential of combining an adaptive ratio approach with our method, we adopt a simple heuristic: we increase the parameter ratios for the key and value projection matrices while reducing them for the query and output projection matrices. This simple combination yields a consistent gain, and we name this variant as SLIM+. We list the details on the hyperparameter in Appendix A.3.

Table 1: Experimental results on Mistral-7B across different parameter ratios.

| Ratio | Method | MSE ↓ | Perplexity ↓ | Classification Accuracy ↑ | | | | | | | Generation ↑ | |
|---|---|---|---|---|---|---|---|---|---|---|---|---|
| | | | WikiText-2 | Openb. | ArcE. | WinoG. | HellaS. | PIQA | MathQA | Avg. | TruthQA | GSM8k |
| 1.0 | Original | 0.00 | 5.32 | 34.60 | 80.26 | 74.90 | 61.48 | 80.63 | 36.08 | 61.33 | 40.76 | 38.29 |
| 0.8 | FWSVD | 0.79 | 25.49 | 24.20 | 64.69 | 65.98 | 38.72 | 70.24 | 22.08 | 47.65 | 25.95 | 0.30 |
| | ASVD | 0.93 | 21.38 | 21.40 | 57.03 | 59.12 | 36.22 | 67.46 | 24.76 | 44.33 | 24.11 | 1.14 |
| | SVD-LLM | 0.46 | 7.62 | 27.20 | 68.94 | 67.72 | 41.42 | 69.37 | 28.74 | 50.57 | 34.64 | 4.32 |
| | AdaSVD | 0.43 | 7.62 | 27.20 | 68.48 | 67.72 | 41.61 | 69.48 | 28.91 | 50.57 | 33.17 | 3.79 |
| | SLIM | **0.25** | 6.73 | 31.80 | 74.92 | 69.46 | 51.87 | 75.35 | 31.93 | 55.89 | 41.74 | 12.43 |
| | SLIM+ | 0.28 | **6.68** | **32.20** | **75.93** | **70.40** | **53.24** | **75.95** | **32.33** | **56.68** | **45.90** | **13.04** |
| 0.6 | FWSVD | 0.99 | 238.09 | 14.40 | 34.01 | 53.91 | 27.83 | 54.68 | 19.50 | 34.06 | 13.95 | 0.23 |
| | ASVD | 1.20 | 2900.55 | 13.40 | 27.15 | 49.57 | 26.06 | 51.31 | 21.68 | 31.53 | 13.34 | 0.08 |
| | SVD-LLM | 0.69 | 16.14 | 19.40 | 42.63 | 55.41 | 30.45 | 58.00 | 22.38 | 38.05 | 3.06 | 1.82 |
| | AdaSVD | 0.66 | 15.64 | 19.40 | 42.85 | 59.67 | 31.81 | 59.79 | 22.38 | 39.32 | 7.71 | 1.14 |
| | SLIM | **0.37** | 9.20 | 25.80 | 64.48 | **64.33** | 41.11 | 67.30 | 26.87 | 48.32 | 30.84 | **2.20** |
| | SLIM+ | **0.37** | **9.05** | **27.20** | **67.42** | 64.17 | **42.84** | **68.72** | **27.10** | **49.58** | **32.44** | 2.12 |
| 0.4 | FWSVD | 1.02 | 1528.48 | 12.80 | 26.98 | 48.70 | 25.76 | 52.45 | 18.63 | 30.89 | 19.46 | 0.00 |
| | ASVD | 1.12 | 4581.07 | 14.80 | 25.38 | 48.78 | 25.86 | 51.58 | 19.20 | 30.93 | **29.38** | 0.00 |
| | SVD-LLM | 0.92 | 87.25 | 12.20 | 28.07 | 49.88 | 26.46 | 52.29 | 20.37 | 31.55 | 1.59 | 0.23 |
| | AdaSVD | 0.89 | 70.11 | 13.20 | 28.28 | 52.80 | 26.49 | 52.07 | 20.27 | 32.19 | 1.10 | 0.15 |
| | SLIM | **0.47** | 18.87 | **19.00** | 42.47 | **56.20** | 31.59 | 57.24 | 22.24 | 38.12 | 12.73 | 1.29 |
| | SLIM+ | **0.47** | **18.29** | 18.80 | **45.29** | 55.80 | **31.75** | **58.00** | **22.68** | **38.72** | 8.08 | **2.20** |
| 0.2 | FWSVD | 1.05 | 4074.49 | 13.60 | 26.73 | 49.64 | 25.44 | 51.63 | 18.19 | 30.87 | 22.03 | 0.00 |
| | ASVD | 1.63 | 7671.85 | 14.20 | 26.35 | 51.22 | 25.55 | 51.36 | 17.76 | 31.07 | 23.99 | 0.00 |
| | SVD-LLM | 1.01 | 1062.89 | 13.20 | 27.10 | 49.33 | 25.80 | 52.39 | 20.60 | 31.40 | 0.00 | 0.00 |
| | AdaSVD | 0.99 | 708.04 | 13.20 | 26.94 | 47.99 | 25.91 | 51.52 | 21.07 | 31.11 | 0.00 | 0.00 |
| | SLIM | 0.59 | 105.24 | **15.60** | 28.96 | 49.64 | 26.12 | **53.75** | 20.74 | 32.47 | 22.77 | 0.53 |
| | SLIM+ | **0.58** | **90.02** | 15.00 | **29.88** | **51.78** | **26.56** | 53.65 | **20.77** | **32.94** | **25.58** | **0.91** |

**Implementations.** For SLIM and SLIM+, unless otherwise specified, we conduct 10 epochs of gradient-based optimization on the block MSE objective (equation 4) with Adam (batch_size=1, $lr = 1e - 7$), as we observe that a smaller batch size seems to perform more consistently across different settings, which is also seen by Marek et al. (2025). We set $\eta_0 = 1$ for the regularization parameter.

The results without such optimization are shown in Table 3. All experiments are conducted with PyTorch and Huggingface Transformers on a single NVIDIA A100-40G GPU.

**Evaluation Protocols.** We evaluate the performance on 6 open-sourced LLM pre-trained models: gemma-3-1b (GemmaTeam, 2025), gemma-3-4b (GemmaTeam, 2025), Qwen3-8B (Yang et al., 2025), OPT-6.7B (Zhang et al., 2022), DeepSeek-llm-7b (Bi et al., 2024), and Mistral-7B (Jiang et al., 2024). We follow the evaluation setup of recent low-rank compression methods, such as ASVD (Yuan et al., 2023), SVD-LLM (Wang et al., 2025), and AdaSVD (Li et al., 2025). Specifically, for the calibration set, we randomly select 256 samples from the WikiText-2 (Merity et al., 2016) to conduct model compression. For evaluation tasks, we use the LM-Evaluation-Harness Framework (Gao et al., 2021) which includes six classification datasets (OpenbookQA (Mihaylov et al., 2018), WinoGrande (Sakaguchi et al., 2021), HellaSwag (Zellers et al., 2019), ArcE (Clark et al., 2018), PIQA (Bisk et al., 2020), MathQA (Amini et al., 2019)) and two generation datasets (TruthfulQA (Lin et al., 2021) and GSM8k (Cobbe et al., 2021)).

For different metrics, we use ↑ to indicate the higher the better and ↓ to indicate the lower the better.

## 5.2 MAIN RESULTS

We evaluated our proposed methods on several language models with different parameter ratios, with the results summarized in Table 1.

Even with a fixed parameter ratio across all weight matrices, our method (SLIM) significantly outperforms other low-rank compression baselines. By considering the structure of the Transformer block, rather than individual matrices, our method has a substantial reduction in the normalized Mean Squared Error (MSE), $\|\bar{f}^l(\bar{X}^l) - f^l(X^l)\|_F^2 / \|f^l(X^l)\|_F^2$. (We only listed the MSE for the last layer. See Figure 1 for the comparison of MSE across different layers.) This reduced MSE also

Table 2: Experimental results across different large language models at parameter ratio 0.8 with Perplexity on WikiText-2 and the average accuracy for the six classification tasks.

| Method | Gemma-3-1B | Gemma-3-4B | OPT-6.7B | DeepSeek-7B | Qwen3-8B |
|--------|------------|------------|----------|-------------|----------|
| | WikiText-2 Perplexity ↓ / Classification Accuracy ↑ | | | | |
| Original | 10.58 / 51.18 | 7.39 / 60.34 | 10.90 / 51.54 | 6.89 / 57.16 | 9.79 / 61.05 |
| SVD-LLM | 51.29 / 34.97 | 15.40 / 43.09 | 12.11 / 50.00 | 9.20 / 49.35 | 14.02 / 51.16 |
| AdaSVD | 22.06 / 39.39 | 15.40 / 44.01 | 11.81 / **50.40** | 9.20 / 49.14 | 13.59 / 51.60 |
| SLIM+ | **19.47 / 41.96** | **11.62 / 49.03** | **11.60** / 50.40 | **8.50 / 52.40** | **14.24 / 54.03** |

Table 3: Comparison of the baselines and our proposed method with no gradient-based optimization (OPT) on the block MSE objective (equation 4).

| Method | Param Rate 0.8 | Param Rate 0.6 | Param Rate 0.4 | Param Rate 0.2 |
|--------|----------------|----------------|----------------|----------------|
| | WikiText-2 Perplexity ↓ / Classification Accuracy ↑ | | | |
| SVD-LLM | 7.62 / 50.57 | 16.14 / 38.05 | 87.25 / 31.55 | 1062.89 / 31.40 |
| AdaSVD | 7.62 / 50.57 | 15.40 / 39.32 | 70.11 / 32.19 | 708.04 / 31.11 |
| SLIM+ (no OPT) | **6.98 / 54.15** | **9.49 / 43.89** | **18.87 / 35.17** | **74.63 / 32.65** |

translates to lower perplexity and improved accuracy on downstream classification and generation benchmarks.

In particular, SLIM improves classification accuracy by more than 5% at parameter ratios between 0.4 and 0.8 and increases the precision of GSM8K by nearly 10% at a 0.8 ratio. Furthermore, performance can be enhanced by incorporating adaptive rank selection (SLIM+). This demonstrates the compatibility of our method with adaptive rank selection.

We further conduct the experiments on multiple LLMs. Table 2 shows that our method outperforms the baselines in large-language models of different models of different families and sizes.

## 5.3 EFFECTIVENESS OF CLOSED-FORM SOLUTION OBTAINED BY SLIM

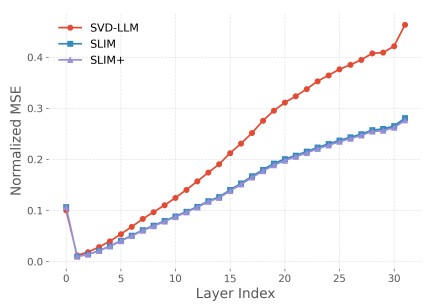

Figure 1: The normalized MSE of each layer for Mistral-7B at parameter ratio 0.8.

We conducted two ablation studies to demonstrate the effectiveness of the closed-form solution obtained by SLIM. First, we evaluated SLIM without any finetuning on the block MSE objective. As shown in Table 3, this approach already outperforms the baselines, highlighting the strength of the solution itself. Conversely, we tested a scenario with only finetuning on SVD-LLM, omitting our closed-form solution. Figure 2 shows that finetuning alone is insufficient for the model to reach the performance achieved by our full method. Together, these observations confirm that our proposed closed-form solution provides a significant and decisive advantage.

## 5.4 COMPARISON UNDER THE INSTRUCTION FINETUNING SETTING

SVD-LLM proposed to apply instruction finetuning after model compression. We follow the same setup and compare the results with SVD-LLM and AdaSVD. Table 4 shows the result with instruction finetuning on the entire alpaca-cleaned dataset of 51k samples.

As we can see, under the instruction finetuning setting, our proposed method still outperforms the baselines across different parameter ratios.

Table 4: Experimental results with instruction finetuning on alpaca-cleaned dataset.

| Method | Param Rate 0.8 | Param Rate 0.6 | Param Rate 0.4 | Param Rate 0.2 |
|---|---|---|---|---|
| | WikiText-2 Perplexity ↓ / Classification Accuracy ↑ | | | |
| SVD-LLM (IFT) | 7.99 / 54.06 | 10.10 / 48.75 | 15.89 / 40.14 | 48.18 / 34.61 |
| AdaSVD (IFT) | 7.99 / 53.71 | 10.58 / 49.14 | 16.14 / 40.90 | 45.26 / 34.53 |
| SLIM+ (IFT) | 7.74 / 56.15 | 7.99 / 51.95 | 12.37 / 44.26 | 28.33 / 35.62 |

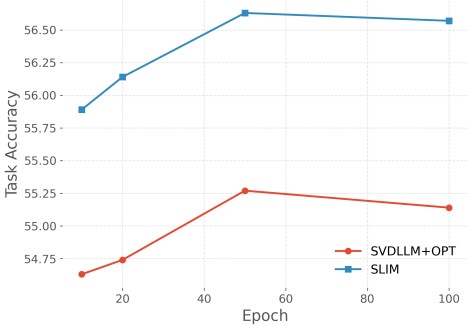

Figure 2: Comparing SLIM and SVD-LLM combined with different number of epochs of block MSE optimization.

Figure 3: Comparing SLIM and SVD-LLM compressed on different numbers of calibration samples.

## 5.5 COMPARISON WITH PRUNING METHODS

We compare our method against two model pruning baselines: BlockPruner (Zhong et al., 2024) and LLM-Pruner (Ma et al., 2023). Due to the high memory requirements of LLM-Pruner, we evaluated it on a smaller calibration set (64 samples of length 128). As shown in Table 5, our method outperforms both baselines in their respective settings. In particular, our approach using the smaller dataset (S=64, L=128) still surpasses the performance of BlockPruner, which was calibrated on a much larger dataset (S=256, L=2048).

Table 6: Experiments for combination with quantization.

| Method | Param Rate 0.8 | Param Rate 0.6 | Param Rate 0.4 | Param Rate 0.2 |
|---|---|---|---|---|
| | WikiText-2 Perplexity ↓ / Classification Accuracy ↑ | | | |
| SVDLLM (align 128) | 7.39 / 52.02 | 15.40 / 38.68 | 72.33 / 32.03 | 938.00 / 31.50 |
| SVDLLM (align 128, awq) | 7.51 / 51.93 | 15.64 / 38.52 | 72.33 / 32.16 | 1096.63 / 31.72 |
| SLIM (align 128) | 6.68 / 56.98 | 9.20 / 48.74 | 17.73 / 39.07 | 81.96 / 32.85 |
| SLIM (align 128, awq) | 6.78 / 56.52 | 9.34 / 48.58 | 18.00 / 38.85 | 84.56 / 32.72 |

## 5.6 COMBINATION WITH QUANTIZATION.

Table 6 shows the result of combining low-rank compression methods with AWQ quantization to 4 bits. To employ AWQ quantization, one need to align the dimensions of the matrices to be the multiples of a certain number. We follow the setting of ASVD (Yuan et al., 2023) to round up the ranks to multiples of 128. As we can see, our proposed method can be combined with quantization to further reduce the model size with little loss in accuracy.

Table 5: Comparing SLIM with other model pruning methods.

| Method | PPL / Accuracy |
|---|---|
| LLM-Pruner (S=64,L=128) | 7.51 / 48.70 |
| BlockPruner (S=256,L=2048) | 9.13 / 50.20 |
| SLIM (S=64,L=128) | 8.64 / 52.04 |
| SLIM (S=256,L=2048) | 6.73 / 55.89 |

## 5.7 ABLATION STUDY

For the ablation study, we first investigate the impact of the number of calibration samples. As shown in Figure 3, both SVD-LLM and our method benefit from increasing the sample size. However, our approach demonstrates superior data efficiency, outperforming SVD-LLM even when using just 64 samples.

Next, we ablated the core mechanisms the SLIM method. The results in Table 7 confirm that both the structural correction and the original weight regularization are critical, as removing either component significantly degrades performance.

Table 7: Ablation of SLIM.

| Method | PPL / Accuracy |
|---|---|
| SLIM (no OPT) | 6.94 / 52.80 |
| SLIM (no OPT, w/o structural-correction) | 7.39 / 50.57 |
| SLIM (no OPT, w/ vanilla regularization) | 7.00 / 51.88 |

## 5.8 QUALITATIVE RESULTS

Here, we provide some qualitative results to demonstrate the difference between our method and SVD-LLM. Observing that our proposed method has a much higher accuracy than SVD-LLM on GSM8K at parameter ratio 0.8, we sampled 50 GSM8K problems and examined the output for SVD-LLM and our proposed method. We highlight the final extracted answer in yellow.

Table 8: Comparison of the output of SVD-LLM and SLIM on a GSM8K example

| SVD-LLM | SLIM |
|---|---|
| **Answer:** Eliza worked 40 hours at $10 per hour. She worked 5 hours at 1.2 times her regular hourly rate. She worked 40 hours at $10 per hour and 5 hours at 1.2 times her regular hourly rate. She worked 40 hours at $10 per hour and 5 hours at 1.2 times her regular hourly rate. She worked 40 hours at $10 per hour and 5 hours at 1.2 times her regular hourly rate ... | **Answer:** Eliza worked for 40 hours at $10 per hour. She worked for 5 hours at 1.2 times her regular hourly rate. So she worked for 40 hours at $10 per hour and 5 hours at 1.2 times $10 per hour. 40 x $10 = <<40 * 10 = 400 >> 400 dollars for 40 hours at $10 per hour. 5 x 1.2 x $10 = <<5 * 1.2 * 10 = 60 >> 60 dollars for 5 hours at 1.2 times $10 per hour. Altogether, Eliza earned 400 + 60 = <<400 + 60 = 460 >> 460 dollars for the week. ; ; ; ; ; ... |
| Incorrect | Correct |

A representative example in Table 8 highlights a common failure mode for SVD-LLM: getting stuck in repetitive loops. In contrast, our method better preserves the original model's mathematical reasoning abilities (see Appendix A.4 for more examples). Interestingly, in the two cases where SVD-LLM produced a correct final answer while ours was incorrect, we found that SVD-LLM's underlying reasoning was flawed.

## 6 CONCLUSION

In this paper, we introduces SLIM, a new method for the low-rank compression of large language models. We formulate a joint optimization problem to find the optimal low-rank weights for an entire transformer block, thereby minimizing the output reconstruction error. Our formulation allows the incorporation of key architectural elements, including residual connections and normalizations. Experimental results demonstrate that our method consistently achieves task accuracy improvements of over 5% compared to existing techniques across a range of compression ratios and model families.

## REPRODUCIBILITY STATEMENT

We are dedicated to guaranteeing the reproducibility of our work. The proposed SLIM method is described in Section 3, with detailed pseudocode provided in Algorithm 1. All hyperparameters, benchmark details, evaluation metrics, and baselines are specified in the experimental setup (Section 5.1). Upon acceptance of this paper, we will publicly release our source code.

## ETHICS STATEMENT

This paper focuses on model compression techniques. No direct ethical concerns or issues arise from the technical methodology presented.

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

# A APPENDIX

## A.1 USE OF LLMS

We used large language models (LLMs) solely for the purpose of polishing the writing in this paper.

## A.2 PROOF OF THEOREM 1

By our definition

$$\hat{X} = \text{concat}([X, \eta^{1/2}I]), \hat{Y} = \text{concat}([Y, \eta^{1/2}I]),$$

thus we have

$$\|\bar{W}_{\bar{h}}X - Y\|_F^2 + \eta\|\bar{W}_{\bar{h}} - W_h\|_F^2 = \|\bar{W}_{\bar{h}}X - Y\|_F^2 + \|\eta^{1/2}\bar{W}_{\bar{h}}I - \eta^{1/2}W_hI\|_F^2$$
$$= \|\bar{W}_{\bar{h}}\text{concat}([X, \eta^{1/2}I]) - \text{concat}([Y, \eta^{1/2}I])\|_F^2 = \|\bar{W}_{\bar{h}}\hat{X} - \hat{Y}\|_F^2. \tag{12}$$

Theorem 1 from Izenman (1975) states that for the reduced-rank linear regression problem

$$\min_{\bar{W}_{\bar{h}};\text{rank}(\bar{W}_{\bar{h}})\leq r}\|\bar{W}_{\bar{h}}X - Y\|_F^2, \tag{13}$$

the solution is

$$\bar{W}_{\bar{h}} = \text{SVD}_r(YX^\dagger X)X^\dagger. \tag{14}$$

Consequently, by change of variables, for

$$\min_{\bar{W}_{\bar{h}};\text{rank}(\bar{W}_{\bar{h}})\leq r}\|\bar{W}_{\bar{h}}\hat{X} - \hat{Y}\|_F^2, \tag{15}$$

the solution should be

$$\bar{W}_{\bar{h}} = \text{SVD}_r((\hat{Y}\hat{X}^\dagger)\hat{X})\hat{X}^\dagger, \tag{16}$$

which completes the proof.

## A.3 HYPER PARAMETER FOR SLIM+

The only extra hyper parameter of SLIM+ compared to SLIM is the parameter ratio for the key and value projection matrix. We reduce the parameter ratio for the query and output projection matrix to keep the same total parameter ratio. Since the query and output projection matrix is often larger than the key and value projection matrix, their reduction is often much smaller than the increment of the key and value projection matrix. For Mistral-7B, we set the parameter ratio to 1.0 for the key and value projection matrix for total parameter ratio 0.8, 0.6, and 0.4. For total parameter ratio 0.2, the parameter ratio for the key and value projection matrix is set to 0.4. For other LLM models, we only experiment with total parameter ratio 0.8, and the parameter ratio for the key and value projection matrix is set to 1.0.

## A.4 ADDITIONAL GSM8K EXAMPLES

We sampled 50 problems of GSM8K and example the output for SVD-LLM and our proposed method. In Table 9, we've included five examples for a closer look. We highlight the final extracted answer in yellow. The first three examples show instances where SLIM answered correctly, while SVD-LLM failed. Conversely, in the remaining two examples, SVD-LLM provided the correct final answer. Interestingly, in this two examples, we found that SVD-LLM's underlying reasoning was flawed.

Table 9: Comparison of the outputs of SVD-LLM and SLIM on GSM8K

| SVD-LLM | SLIM |
|---------|------|

**Question:** Eliza's rate per hour for the first 40 hours she works each week is $10. She also receives an overtime pay of 1.2 times her regular hourly rate. If Eliza worked for 45 hours this week, how much are her earnings for this week?

**Answer:** Eliza worked 40 hours at $10 per hour. She worked 5 hours at 1.2 times her regular hourly rate. She worked 40 hours at $10 per hour and 5 hours at 1.2 times her regular hourly rate. She worked 40 hours at $10 per hour and 5 hours at 1.2 times her regular hourly rate. She worked 40 hours at $10 per hour and 5 hours at 1.2 times her regular hourly rate . . .

Incorrect

**Question:** Eliza's rate per hour for the first 40 hours she works each week is $10. She also receives an overtime pay of 1.2 times her regular hourly rate. If Eliza worked for 45 hours this week, how much are her earnings for this week?

**Answer:** Eliza worked for 40 hours at $10 per hour. She worked for 5 hours at 1.2 times her regular hourly rate. So she worked for 40 hours at $10 per hour and 5 hours at 1.2 times $10 per hour. 40 x $10 = <<40 * 10 = 400 >> 400 dollars for 40 hours at $10 per hour. 5 x 1.2 x $10 = <<5 * 1.2 * 10 = 60 >> 60 dollars for 5 hours at 1.2 times $10 per hour. Altogether, Eliza earned 400 + 60 = <<400 + 60 = 460 >> 460 dollars for the week. ; ; ; ; ; . . .

Correct

---

**Question:** John takes care of 10 dogs. Each dog takes .5 hours a day to walk and take care of their business. How many hours a week does he spend taking care of dogs?

**Answer:** John takes care of dogs for .5 hours a day. John takes care of dogs for .5 hours a day * 7 days a week = << .5 * 7 = 3.5 >> 3.5 hours a week. John takes care of dogs for .5 hours a day * 7 days a week = << .5 * 7 = 3.5 >> 3.5 hours a week . . .

Incorrect

**Question:** John takes care of 10 dogs. Each dog takes .5 hours a day to walk and take care of their business. How many hours a week does he spend taking care of dogs?

**Answer:** John spends 10 * .5 = <<10 * .5 = 5 >> 5 hours a day taking care of dogs. John takes care of dogs every day, so he spends 5 * 7 = <<5 * 7 = 35 >> 35 hours a week taking care of dogs. ; ; ; ; ; . . .

Correct

---

**Question:** Grandma Jones baked 5 apple pies for the fireman's luncheon. She cut each pie into 8 pieces and set the five pies out on the buffet table for the guests to serve themselves. At the end of the evening, after the guests had taken and eaten their pieces of pie, there were 14 pieces of pie remaining. How many pieces were taken by the guests?

**Answer:** Grandma Jones baked 5 apple pies. She cut each pie into 8 pieces. She set the five pies out on the buffet table for the guests to serve themselves. At the end of the evening, after the guests had taken and eaten their pieces of pie, there were 14 pieces of pie remaining. 14 pieces were taken by the guests. ; ; ; ; ; . . .

Incorrect

**Question:** Grandma Jones baked 5 apple pies for the fireman's luncheon. She cut each pie into 8 pieces and set the five pies out on the buffet table for the guests to serve themselves. At the end of the evening, after the guests had taken and eaten their pieces of pie, there were 14 pieces of pie remaining. How many pieces were taken by the guests?

**Answer:** Grandma Jones baked 5 apple pies. She cut each pie into 8 pieces. So she cut 40 pieces of pie. She set 5 pies out on the buffet table. So there were 40 pieces of pie on the buffet table. At the end of the evening, after the guests had taken and eaten their pieces of pie, there were 14 pieces of pie remaining. So 26 pieces of pie were taken by the guests. ; ; ; ; ; . . .

Correct

**Question:** Marissa is hiking a 12-mile trail. She took 1 hour to walk the first 4 miles, then another hour to walk the next two miles. If she wants her average speed to be 4 miles per hour, what speed (in miles per hour) does she need to walk the remaining distance?

**Answer:** Marissa walked 4 miles in 1 hour. She walked 2 miles in 1 hour. Marissa walked 4 miles + 2 miles = << 4 + 2 = 6 >> 6 miles in 2 hours. Marissa walked 6 miles in 2 hours. Marissa walked 6 miles in . . .

Correct, but the reasoning is flawed

**Question:** Marissa is hiking a 12-mile trail. She took 1 hour to walk the first 4 miles, then another hour to walk the next two miles. If she wants her average speed to be 4 miles per hour, what speed (in miles per hour) does she need to walk the remaining distance?

**Answer:** Marissa walked 4 miles in 1 hour, so her average speed was 4 miles per hour. She walked another 2 miles in another hour, so her average speed was still 4 miles per hour. She needs to walk the remaining 6 miles in 2 hours so that her average speed will be 4 miles per hour. She needs to walk 6 miles in 2 hours so that her average speed will be 4 miles per hour. . . .

Incorrect

**Question:** Charlie wants to sell beeswax candles. For every pound of beeswax, he can make 10 tapered candles. One pound of beeswax and the wicks cost $10.00 in supplies. If he sells each candle for $2.00 each, what is his net profit if he makes and sells 20 candles?

**Answer:** Charlie sells 20 candles. He sells 20 candles for $2.00 each. He sells 20 candles for $2.00 each. He sells 20 cand . . .

Correct, but the reasoning is flawed

**Question:** Charlie wants to sell beeswax candles. For every pound of beeswax, he can make 10 tapered candles. One pound of beeswax and the wicks cost $10.00 in supplies. If he sells each candle for $2.00 each, what is his net profit if he makes and sells 20 candles?

**Answer:** Charlie can make 10 tapered candles from one pound of beeswax. One pound of beeswax costs $10.00 in supplies. Charlie sells each candle for $2.00 each. Charlie makes and sells 20 candles. Charlie's net profit is $200.00.\n ; ; ; ; ; . . .

Incorrect

