# OpenReview forum: "SLIM: Structure-aware Low-rank Inference Model"
_ICLR.cc/2026/Conference — Submitted to ICLR 2026_

### Official Review · Reviewer_BECz · 2025-10-27

**Soundness:** 3
**Presentation:** 2
**Contribution:** 3
**Rating:** 4
**Confidence:** 3

**Summary:**

This paper takes the propagation of low-rank approximation error in transformer and proposes to sequentially update each weight matrix.

Specifically, this paper shows each step reduces to solving a regularized rank-constrained linear regression problem with a closed-form solution and also shows that a gradient-descent–based refinement can further enhance the performance.

**Strengths:**

1. This paper brings an insight into the low-rank compression for neural networks that have sequential structures. It introduces a sequential optimization at the transformer block level.
2. SLIM is compatible with many other compression techniques, such as pruning and quantization.
3. The gain obtained on pretrained model can be maintained on instruction following fine-tuning.

**Weaknesses:**

1. The value of the close-form solution is undermined by the gradient-based solution. See Questions 5 and 6.
2. Experiment setups are unclear, which undermines the reproducibility. See Questions 1, 2, and 3,

**Questions:**

1. What sample data did you use for the low-rank approximation? How did you choose it and why did you use it?
2. What is the initialization of the low-rank weights in gradient-based optimization of Equation (4)? Is it random, truncated SVD, or the close-form solution?
3. For the 10 epochs, did you repeat {optimize Block 1, ..., optimize Block N} for 10 times or optimize Block 1 for 10 times and then optimize Block 2 for 10 times, ...?
4. What is the MSE of SLIM (no OPT)?
5. What is the computational complexity of the gradient-based optimization of Equation (4)? It should be $O(d^2)$?
6. If the complexity of gradient-based algorithm is $O(d^2)$, what is the value or insight of the close form derivation given the close-form algorithm is much more complex and performs worse than the gradient-based algorithm?

---

> ### Author Response · Authors · 2025-11-28
>
> Dear Reviewer BECz,
>
> We thank the reviewer for the feedback.
>
> > What sample data did you use for the low-rank approximation? How did you choose it and why did you use it?
>
> As shown in the EvaluationProtocols section of Sec 5.1, we follow the previous work (ASVD, AdaSVD, and SVDLLM) to randomly select 256 samples from the WikiText-2 for the low-rank approximation.
>
> > What is the initialization of the low-rank weights in gradient-based optimization of Equation (4)? Is it random, truncated SVD, or the close-form solution?
>
> For our method, we use our close-form solution for initialization. In Figure 2, we compare this with initializing with SVDLLM. As shown in Figure 2, we observe that the our close-form solution provides a much better initialization than existing baselines like SVD-LLM,
>
> > For the 10 epochs, did you repeat {optimize Block 1, ..., optimize Block N} for 10 times or optimize Block 1 for 10 times and then optimize Block 2 for 10 times, ...?
>
> We optimize Block 1 for 10 times and then we compress and optimize Block 2 for 10 times. Consequently, we are considering the output of the optimized Block 1 when compressing Block 2.
> > What is the MSE of SLIM (no OPT)?
>
> As we can see in the following table, even with no OPT, the relative MSE of SLIM is still much lower than SVDLLM and has higher accuracies on the six classification tasks.
> | Ratio | Method | Relative MSE | ClassificationAcc. |
> |---|---|---|---|
> | 0.8 | SVDLLM | 0.46| 50.57|
> | |SLIM (no OPT) | 0.27| 52.80|
> | |SLIM (10 epoch) |0.25| 55.89|
> | |SLIM (20 epoch)|0.27|56.14|
> ||SLIM (50 epoch)|0.25|56.63|
> ||SLIM (100 epoch)|0.24|56.57|
>
> > What is the computational complexity of the gradient-based optimization of Equation (4)?
>
> The computational complexity of the gradient-based optimization is $O(nd^2)$, where $n$ is the total number of tokens. In general $n$ should be much larger than $d$ to provide enough information for accurate compression (In our setting, $n=2048*256, d=14336$).
>
> Consequently, $O(nd^2)$ should be the dominating term in complexity as it is larger than $O(d^3)$.
>
> > The value of the close-form solution is undermined by the gradient-based solution.
> > what is the value or insight of the close form derivation
>
> As mentioned earlier, the complexity of the close-form solution is similar to that of the gradient-based solution.
>
> Our proposed method **combines the close-form solution and the gradient-based solution**. In Figure 2, we show that our close-form solution provides a much better initialization than existing baselines like SVD-LLM (which has the same computational complexity as our close-form solution). Thus, both the close-form solution and the gradient-based solution are important parts of our proposed method.

---

### Official Review · Reviewer_aVy8 · 2025-10-27

**Soundness:** 2
**Presentation:** 2
**Contribution:** 2
**Rating:** 2
**Confidence:** 4

**Summary:**

This paper studies low rank compression of LLMs. They propose a new method based on reduced rank regression, and provide numerical experiments for different LLMs with up to 7B parameters. The proposed method performs well compared to the baselines such as SVD-LLM.

**Strengths:**

The numerical experiments appear to be reasonable. There is a diverse choice of models and benchmarks, though some ablations are missing.

**Weaknesses:**

I think the paper has very limited technical contribution.

- Handling of residuals and normalization: The paper's main claim is that it performs compression based on the whole transformer block instead of a layerwise scheme. However, this claim is not really supported by the paper. In particular, under the sequential compression that this paper studies, the compression happens for one layer at a time, which is just layerwise. Additionally, the residual connections end up as a bias term that does not change the problem fundamentally, and the normalization terms are just approximated out of existence, I believe simplifying it to the choice of $\eta$.

- The effect of $\eta$ is not studied as far as I can tell. Given that normalization is just tuning $\eta$, I think some investigation is required here.

- Comparison with pruning methods: I would like to ask the authors to compare with additional pruning methods, given that LLM-Pruner has memory issues. For example, https://arxiv.org/abs/2406.07831.

**Questions:**

- Could the author please provide the ablations requested above?

- What happens in terms of quality if one just uses reduced rank regression directly without considering normalization and residual connections? Could the authors please provide some numerical results?

---

> ### Author Response · Authors · 2025-11-28
>
> Dear Reviewer aVy8,
>
> We thank the reviewer for the feedback, but we would like to make some clarifications.
>
> > The paper's main claim is that it performs compression based on the whole transformer block instead of a layerwise scheme. However, the compression happens for one layer at a time, which is just layerwise
>
> While the compression happens for one layer at a time, we adopt the output of the compressed model instead of the full model (as in the baselines). Additionally, for certain linear layers, we consider its joint effect with other crucial layers in the block like the residual connection and layer normalization. These two mechanisms allow us to **account for the error accumulation within the entire transformer block**, which is ignored by the prior works during the compression step.
>
> As shown in Table1, this results in our proposed method having a much stronger performance than the baselines.
> > Additionally, the residual connections end up as a bias term that does not change the problem fundamentally.
>
> While the consideration of residual connections still results in a rank-constrained linear regression problem for W, it does change the solution and has a huge impact on the performance.
>
> We would like to point out that **the inclusion of residual connections is crucial** for mitigating accumulation error. Otherwise, even though one considers the error during the attention and the mlp layer, the error will still keep accumulating across layers through the residual connection.
>
> Empirically, this is also verified in our ablation study in Sec 5.7, which shows that when we remove the structural correction we see little improvement over SVDLLM.
>
> | Ratio | Method | WikiText-2 | Openb. |  ArcE. |  WinoG. |  HellaS.  |  PIQA  |  MathQA  |  Avg. |
> |---|---|---|---|---|---|---|---|---|---|
> | 0.8 | SVDLLM | 7.62 | 27.20 | 68.94 | 67.72 | 41.42 | 69.37 | 28.74 | 50.57 |
> | | SLIM (no OPT, wo/ residual) | 7.39 | 25.60|	69.44|	67.80|	43.58|	68.44|	28.58|	50.57|
> | | SLIM (no OPT)| 6.94 | 29.60|	71.97|	69.06|	45.91|	70.62|	29.61|	52.80|
>
> > the normalization terms are just approximated out of existence, I believe simplifying it to the choice of $\eta$
>
> In our paper, $\eta$ controls the strength of regularization for the compressed parameters. The consideration of the normalization terms results in having individual weights to each token inversely proportional to its output length. This is different from tuning the regularization parameter $\eta$.
>
> > What happens in terms of quality if one just uses reduced rank regression directly without considering normalization and residual connections? Could the authors please provide some numerical results?
>
> As shown in our ablation study in Sec 5.7 and the table above, when we remove the structural correction we see little improvement over SVDLLM. This stresses the importance of taking these important structures, including normalization and residual connections, into account during compression.

---

### Official Review · Reviewer_KGra · 2025-10-31

**Soundness:** 3
**Presentation:** 3
**Contribution:** 3
**Rating:** 6
**Confidence:** 2

**Summary:**

This paper optimizes the traditional low-rank decomposition methods by jointly optimizing it across the entire transformer block. The author constructs a structure-aware optimization algorithm that consider the residual connections and RMS Norms in the optimization. Experiments across different models and datasets shows SLIM can outperform the baselines over different settings.

**Strengths:**

- This paper offers a novel perspective that jointly optimize the SVD across after an entire transformer block. This idea is intuitive and clearly presented in the paper.
- Wide ablation studies and analysis. Experiments have shown great potentials for this methods to be applied together with other methods, such as quantization.
- Structure-aware methods. As shown in the ablation, this methods respects the residual connections and RMS-Norm of the transformer model, which is novel and critical.

**Weaknesses:**

- Only Wikitext-2 experiment on other model families. Wikitext-2 is a very easy task, while you have included strong models that can solve complicated math and reasoning tasks. I wonder whether you can provide more experiments on other more challenging tasks.
- The greedy algorithm undermines the goal of join optimization and how far are we from achieving the global optimization>

**Questions:**

- Can't we just align the f(x) where f(x) is each sub-components? I feel like optimizing across an entire layer is equivalent to optimizing across each sub-layer?

---

> ### Author Response · Authors · 2025-11-28
>
> Dear Reviewer KGra,
>
> We thank the reviewer for the positive feedback!
>
> > Only Wikitext-2 experiment on other model families.
>
> In Table 2, we not only show the Wikitext-2 results but also the average accuracy across the other six classification tasks (Openb., ArcE., WinoG., HellaS., PIQA, MathQA). We will make this clearer in the paper.
>
> > The greedy algorithm undermines the goal of joint optimization and how far are we from achieving the global optimization
>
> As shown in Figure 2, we observe that the greedy algorithm provides a better initialization than existing baselines like SVD-LLM, while joint optimization can further improve the performance. Thus, both the greedy algorithm and the joint optimization are important parts of our proposed method.
>
> Additionally, the table below shows that both the task accuracy and the relative MSE error is quickly converging, suggesting that we are close to the local optima of the global optimization problem.
>
> | Ratio | Method | Relative MSE | ClassificationAcc. |
> |---|---|---|---|
> | 0.8 | SVDLLM | 0.46| 50.57|
> | |SLIM (no OPT) | 0.27| 52.80|
> | |SLIM (10 epoch) |0.25| 55.89|
> | |SLIM (20 epoch)|0.27|56.14|
> ||SLIM (50 epoch)|0.25|56.63|
> ||SLIM (100 epoch)|0.24|56.57|

---

### Official Review · Reviewer_caDe · 2025-11-01

**Soundness:** 3
**Presentation:** 3
**Contribution:** 1
**Rating:** 2
**Confidence:** 5

**Summary:**

The paper proposes SLIM (Structure-aware Low-rank Inference Model), a new method for compressing large language models by performing joint low-rank optimization across entire transformer blocks instead of compressing each weight matrix independently. SLIM formulates a block-level objective that minimizes output reconstruction error while explicitly accounting for residual connections and normalization layers such as RMSNorm.

**Strengths:**

1. The proposed joint optimization framework explicitly incorporates key architectural elements such as residual connections and normalization.
2. The method achieves consistent and significant accuracy improvements (over 5%) across multiple LLMs and benchmarks under equal compression ratios.

**Weaknesses:**

1. The main weakness of this paper is that its primary contribution—joint optimization of low-rank weights across transformer blocks—has already been explored in prior works [1] and [2]. Both studies consider the cumulative effects of cascading reconstruction errors, and the formulations are highly similar: Equation 6 in [1] and Equation 12 in [2] are nearly identical to Equation 6 in this paper. Furthermore, [1] provides a closed-form solution for low-rank compression, where Equation 11 in this paper closely parallels Equations 4 and 5 in [1]. It seems the authors were not aware of these earlier works, as they are not discussed in the article, which makes the only new contribution the inclusion of residual connections and normalization layers.
2. Lines 237–241 should be revised, as this paper is not the first to introduce the concept of block-level joint optimization for low-rank compression.
3. More advanced low rank compression method [1,3] should be compared.


[1] Zhao, Jialin, Yingtao Zhang, and Carlo Vittorio Cannistraci. "Pivoting Factorization: A Compact Meta Low-Rank Representation of Sparsity for Efficient Inference in Large Language Models." Forty-second International Conference on Machine Learning.

[2] Wei, Jiateng, et al. "Structured optimal brain pruning for large language models." Proceedings of the 2024 Conference on Empirical Methods in Natural Language Processing. 2024.

[3] Wang, Jingcun, et al. "Basis Sharing: Cross-Layer Parameter Sharing for Large Language Model Compression." The Thirteenth International Conference on Learning Representations.

**Questions:**

Check above.

---

> ### Author Response · Authors · 2025-11-28
>
> Dear Reviewer caDe,
>
> We thank the reviewer for the feedback, but we would like to clarify some confusions.
>
> > The main weakness of this paper is that its primary contribution—joint optimization of low-rank weights across transformer blocks—has already been explored in prior works [1] and [2].
> > the only new contribution the inclusion of residual connections and normalization layers
>
> We thank the reviews for pointing us to these papers. **We are aware of the correction mechanism in [1]** as it also appears in the early version (but later removed in the final version) of the SVDLLM paper.
>
> Our mechanism is substantially different from and having several advantages over those in [1] and [2]. First of all, we would like to point out that **the inclusion of residual connections is crucial** for mitigating accumulation error. Otherwise, even though one considers the error during the attention and the mlp layer, the error will still keep accumulating across layers through the residual connection. Empirically, this is also verified in our ablation study in Sec 5.7, which shows that when we remove the structural correction we see little improvement over SVDLLM.
>
> | Ratio | Method | WikiText-2 | Openb. |  ArcE. |  WinoG. |  HellaS.  |  PIQA  |  MathQA  |  Avg. |
> |---|---|---|---|---|---|---|---|---|---|
> | 0.8 | SVDLLM | 7.62 | 27.20 | 68.94 | 67.72 | 41.42 | 69.37 | 28.74 | 50.57 |
> | | SLIM (noOPT, wo/ residual) | 7.39 | 25.60|	69.44|	67.80|	43.58|	68.44|	28.58|	50.57|
> | | SLIM (no OPT)| 6.94 | 29.60|	71.97|	69.06|	45.91|	70.62|	29.61|	52.80|
>
> Second, the correction mechanism in [1] and [2] is separated from their compression step, while the consideration of error accumulation is embedded into our compression step. In [1], the correction is applied after the entire compression process, which means the error accumulation is not considered during their main compression process.
>
> To conduct error correction and compression at the same time, we consider a minimization problem that appears similar yet fundamentally different from the ones in [1] and [2]. In [1], they only solve for U or V one at a time after their entire compression. This not only gives them suboptimal solutions, but also meaning that they are simply solving a normal unconstrained linear regression problem. On the other hand, we are solving U and V at the same time and thus are solving a rank-constrained linear regression problem for W. This fundamentally changes the minimization problem considered. Similarly, in [2], while they apply the correction step before their compression, the process is separate and they are still solving a normal unconstrained linear regression problem.

---

> > ### Author Response · Authors · 2025-11-28
> >
> > > More advanced low rank compression method [1,3] should be compared.
> >
> > The following table shows the result of [1] and [3] on Mistral-7B. (We do not find an open-source implementation of [2].) Additionally, we show the result of SVDLLM with post-compression error correction (appearing in [1] and an earlier version of SVDLLM paper).
> >
> > As one can see, our proposed method **outperform the baselines significantly** at ratio 0.6 and 0.4, as we jointly consider the error accumulation and compression while being aware of the residual connection.
> >
> > PIFA [1] takes any rank-r approximation and reduces its parameter count by $r^2-r$, thus it performs better at larger $r$ and larger parameter ratios like 0.8. This is orthogonal to our contribution and can be easily combined with our method.
> >
> > | Ratio | Method | WikiText-2 | Openb. |  ArcE. |  WinoG. |  HellaS.  |  PIQA  |  MathQA  |  Avg. |
> > |---|---|---|---|---|---|---|---|---|---|
> > | 0.8 | SVDLLM (w/ post-compress corr.) | 7.33 | 25.00|	65.99|	66.93|	41.19|	68.93|	28.44|	49.41|
> > | | PIFA [1] | **5.98** | 30.40|	75.72|	**72.22**|	52.49|	**76.17**|	**33.07**|	**56.68**|
> > | | Basis Sharing [3] | 7.61 | 28.20|	70.50|	67.32|	43.20|	69.70|	29.08|	51.33|
> > | | SLIM+ | 6.68 |**32.20** |**75.93**| 70.40| **53.24**| 75.95| 32.33| **56.68**|
> > | 0.6 | SVDLLM (w/ post-compress corr.) | 12.57 |17.40|	37.96|	57.22|	30.37|	58.38|	22.75|	37.35|
> > | | PIFA [1] |9.64| 20.00|	47.85|	60.93|	33.78|	62.02|	24.42|	41.50|
> > | | Basis Sharing [3] |15.87|17.60|	45.20|	56.99|	31.49|	59.25|	22.61|	38.86|
> > | | SLIM+ | **9.05**| **27.20** |**67.42**| **64.17**| **42.84** |**68.72** |**27.10** |**49.58**|
> > | 0.4 | SVDLLM (w/ post-compress corr.)| 29.22|12.60|	28.28|	52.80|	27.05|	53.54|	20.70|	32.50|
> > | | PIFA [1] | 25.39|13.20|	29.00|	53.43|	27.29|	54.03|	21.04|	33.00|
> > | | Basis Sharing [3] | 65.91|12.00|	29.17|	52.01|	26.98|	52.50|	20.74|	32.23|
> > | | SLIM+ | **18.29** |**18.80**| **45.29**| **55.80**| **31.75**| **58.00**| **22.68** |**38.72**|
> > | 0.2 | SVDLLM (w/ post-compress corr.) | **87.25** |11.20|	26.73|	50.75|	26.25|	52.67|	20.54|	31.36|
> > | | PIFA [1] | **87.25** |12.00|	27.19|	49.96|	26.25|	52.83|	20.13|	31.39|
> > | | Basis Sharing [3] |651.25|12.20|	27.48|	48.54|	25.71|	52.56|	**21.37**|	31.31|
> > | | SLIM+ | 90.02 |**15.00**| **29.88**| **51.78**| **26.56**| **53.65**| 20.77| **32.94**|

---

### Meta-Review · Area_Chair_xnFA · 2026-01-07

**Summary:**

*Paper Summary*: Paper proposes a weight compression approach for transformer architectures that compresses an entire block in distributionally-aware instead of compressing each matrix in isolation and without regard to data distribution, e.g., by performing r-rank approximation using SVD. Main technical contributions are showing that special handling is needed for residual edges and normalization.

Paper is on an important topic, has technical meat, and authors have done an overall good job. Main concerns listed below keep it at borderline leaning. I am currently going with borderline reject, but wont mind if the paper is accepted. The paper should be in really good shape after addressing these concerns.

**Reviewer Concerns:**

See below for detailed list of reviewer concern and how they were addressed. Overall, the main unaddressed concerns are:

1. Norm assumption in handling of normalization is unclear.
2. Lack of evaluation on more domains specially long generation tasks.
3. How does OPT baseline (directly to gradient descent on equation 4) perform? You can do SVD of the original weight matrices, keep just the r-ranks and then do gradient descent to optimize equation 4 for a layer block.

Because of this, I am still leaning towards rejecting this paper but it can be bumped up.

**Reviewer Scores:**

1. Reviewer caDe's main concern is lack of novelty, and previous compression literature either not cited or not compared with. Authors have rebutted by explaining difference with cited work specially pointing that residual connection is important. I agree with this argument and experiments show that it is indeed helpful. I would assume they would have raised their score from 2 to 6.

2. Revewier KGra's main concern are lack of experiments beyond Wikitext-2 for other models and use of greedy initialization. Authors clarified, as is clear in Table 2 label, that other models were also evaluated on all tasks. Authors also clarified that greedy initialization is just for initialization. I think this settles the argument. However, I would have wished for a lot more experiments, specially dataset, given the breadth of LLM datasets out there. For this reason, I find the experiments to be not conclusive yet. I would, therefore, increase the score a bit to 7.

3. Reviewer aVy8's concern are lack of novelty, need for tuning eta and whether added features are relevant. Authors present results showing bias resulting from residual connections is helpful. However, I find the answer about \eta to be unsatisfactory. Firstly, the assumption that $\|\bar{W}_h \bar{g}(\bar{X}^l) \| \approx \|W_h g(X^l)\|$ is not supported nor is it necessary if a gradient based optimization is used to solve equation 8. Further, as the reviewer stated if this assumption is made one gets the loss as $\| \bar{W}_h \bar{g}(\bar{X}^l)  - W_h g(X^l)  \|/c + \eta * Reg$ where if $c$ is guessed then it is no different than tuning $\eta$. However, if it is treated as the original value of $\|W_h g(X^l)\|$ then it is not the case. However, the assumption above still requires ablation. I would increase the score to 4.

4. Reviewer BECz's main concern is if closed form solution is useful, and lack of clarity. Reviewers explain that their solution uses both gradient based and closed form solution treating the latter for initialization. I am assuming that the reason the two work together better is that closed form solution works weight-by-weight and therefore can be sub-optimal. However, I am not able to locate the performance of just performing the gradient based solution on equation 4 without any closed form solution. Authors should clarify it is it covered by some baseline. I would, therefore, keep the score at 4.

---

### Decision · Program_Chairs · 2026-01-26

Reject